



# Relation of aggregate stability and microbial diversity in an incubated sandy soil.

**Frederick Büks[1], Philip Rebensburg[2], Peter Lentzsch[2], Martin Kaupenjohann[1]**

[1] Department of Ecology, Technische Universität Berlin, Germany
[2] Leibniz Center for Agricultural Landscape Research (ZALF e.V.), Müncheberg, Germany






## Abstract

Beside physico-chemical interactions between particulate organic matter (POM), mineral particles and dissolved molecules, microbial biofilms are an important factor of aggregate stability as a proxy of soil quality. Soil primary particles are linked by the highly viscous extracellular biofilm matrix known as extracellular polymeric substance (EPS). Matrix composition depends on the community of biofilm producing species and environmental

conditions affecting gene expression.

This work investigates the influence of microbial biodiversity on soil aggregate stability under controlled environmental conditions. We hypothesized that the formation of different microbial populations would cause different aggregate stabilities. Therefor a sterile sandy agricultural soil with pyrochar amendment from pine wood was incubated for 76 days in

pF-bioreactors. One variant was inoculated with a soil extract, whereas the other one was infected by airborne microbes. A control soil remained unincubated. During the incubation, soil samples were taken for taxon-specific qPCR to determine the abundance of eubacteria, fungi, archaea, acidobacteria, actinobacteria, α-proteobacteria and β-proteobacteria. After incubation soil aggregates were separated for aggregate stability

measurement by ultrasonication, density fractionation and SOC analysis.

As the eubacterial populations of both incubated variants reach a similar level after 49 days, the variant inoculated with the living soil extract shows a much higher fungal population compared to the air-born variant. Within the eubacterial population acidobacteria and β-proteobacteria differ significantly in their abundance between the

variants. Although the variants show a strongly significant difference in eubacterial/fungal population structure, there are only marginal differences in aggregate stability.





# 1 Introduction

Slipping on stones on a river bank, complaining about mucus of a severe cold, marveling about colorful microbial mats in volcanic hot springs and being pleased about the efficiency of biological waste water treatment – we are faced to the wide abundance of biofilms (Costerton et al., 1995). Biofilm formation on surfaces by excretion of extracellular polymeric substance (EPS) is an adaptation behavior of microbial life to manifold environmental stressors (Roberson and Firestone, 1992; Mah and O'Toole, 2001; Weitere et al., 2005; Chang et al., 2007; Ozturk and Aslim, 2010). Hence, in a world of unpleasant ecological conditions the bulk of global bacterial life is supposed to live in biofilms (Davey and O'toole, 2000). As shown in Büks and Kaupenjohann (in revision), surface-bound bacterial life can hold a 45-fold higher bacterial abundance compared with the soil solution. That indicates a dominance of biofilms in soil bacterial communities.

Dependent on environmental conditions and biofilm building species, fresh biofilms have an average water content of 90% with extrema up to 97% (Zhang et al., 1998; Schmitt and Flemming, 1999; Pal and Paul, 2008). Only 10% to 50% of the remaining dry mass are microbial biomass, whereas the remaining matter mainly consists of extracellular macromolecules (More et al., 2014). These are extracellular polysaccharides, proteins, lipids and humic substances with a broad structural diversity within each substance class (Leigh and Coplin, 1992; Allison, 1998; Al-Halbouni et al., 2009; Ras et al., 2011) as well as extracellular DNA (eDNA) (Flemming and Wingender, 2010). Molecular masses of matrix polysaccharides between 13,700 and 2,000,000 Da suggest large strucutral diversity (Votselko et al., 1993).

Polysaccharides, eDNA, proteines and lipids have different functions affecting biofilm stability: Linked with polyvalent cations, polysaccharides and eDNA enhance the viscosity of EPS with increasing concentration and support cell aggregation and adhesion to surfaces (Das et al., 2014). Structural proteins bind extracellular polysaccharides, cell and inanimate surfaces. Also proteinous bacterial pili can take part in macromolecular entanglement. In addition, enzymes play a role in restructuring the biofilm matrix. Lipids on the other hand work as surfactants and support surface colonization and bioavailability of nutrients. (Flemming and Wingender, 2010)

As a result of their mechanical strength (Möhle et al., 2007; Flemming and Wingender, 2010), structure (Van Loosdrecht et al., 2002) and their distribution across the soil aggregate structure (Nunan et al., 2003), biofilms are supposed to play a role in soil



aggregate formation and stabilization (Baldock, 2002). Recent experiments gave evidence for this assumption (Büks and Kaupenjohann, in revision). In addition physico-chemical interactions between organic and mineral primary particles and dissolved molecules play a major role in soil aggregation (Bronick and Lal, 2005).

Although all biofilms contain extracellular polysaccharides, DNA (eDNA), proteins and
lipids as structural agents (Flemming and Wingender, 2010), biofilm composition shows a broad variation. In different single-species biofilms cultivated with each identical media at the same incubation temperatures, specific EPS component compositions are strongly differing (Béjar et al., 1998; Steinberger and Holden, 2005; Celik et al., 2008). But also single-species biofilms of the same organism form EPS of different composition under
varying environmental conditions as demonstrated for *Pseudomonas aeruginosa* (Marty et al., 1992; Ayala-Hernández et al., 2008). That points to a general dependency of EPS composition on species and environmental conditions.

Thus, it seems self-evident that complex multi-species biofilms differing in their phylogenetic diversity should also show a different composition of main biofilm
components – however, experimental proof is still missing. Furthermore, this different EPS composition would generate differing mechanical EPS stability against tractive, compressive and shear forces, which is suggested by the findings of Ayala-Hernández et al. (2008).

Biofilm formation by soil microorganisms mainly appears as a reaction for protection
against ecological stressors as e.g. grazing pressure, toxics and antibiotics, drought and radiation stress, but also act as genetic cross-over hotspot and collective digestive system for diverse soil nutrients (Flemming and Wingender, 2010). On the macro-scale, an environment moderately unpleasant for microorganisms drives biofilm formation and has positive effects on soil structure: High soil aggregate stability results in stable intra-
aggregate fine pores enhancing water retention, whereas the macropores between soil aggregates accelerate water transport, aeration, rootability and decrease erodability and compactability compared to a less aggregated soil (Bennie and Burger, 1988; Bengough and Mullins, 1990; Baumgartl and Horn, 1991; Taylor and Brar, 1991; Ball and Robertson, 1994; Barthes and Roose, 2002; Alaoui et al., 2011). In addition, soil aggregation has
influence on carbon cycling as occluded particulate organic matter (POM) is protected against microbial degradation more effective than free POM (Six et al., 2002; Lützow et al., 2006). These features are identified as properties of fertile agricultural soils, and therefore aggregate stability can be seen as an integral proxy of soil quality.

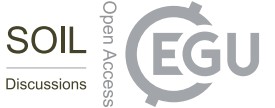

Although research about the microbial and biochemical composition and diversity of single- and multi-species biofilms exists and multiple functions of EPS components are well studied, the relation of microbial community composition and aggregate stability is still unknown. The aim of this work is to do a first step in this field by testing the influence of two fundamentally different microbial populations on the aggregate stability of a sandy agricultural soil. We hypothesized that after establishment of microbial populations different community structures will lead to different aggregate stabilities under further similar conditions. Therefore a gamma-sterilized sandy soil (Su3) with 5% pyrogene biochar amendment from pine wood was inoculated in two variants with microbial and sterile soil extract and incubated for 76 days in a pF-bioreactor under field capacity. Then abundances of eubacteria, fungi, archaea, acidobacteria, actinobacteria, α- and β-proteobacteria DNA were measured using taxon-specific qPCR, and aggregate stability was determined by ultrasonication, density fractioning and carbon analytics.

A significant influence of the microbial populations structure on aggregate stability would confirm that conditions for microbial life should be considered in soil management practices.











## 2 Materials and Methods

### 2.1 Preparation of soil and soil extract

Air-dried soil from a sandy A-horizon (Su3) of an agricultural experimental site in Berge (Germany) was sieved to <2 mm and mechanically disaggregated in a mortar to receive an aggregate-free soil sample. The soil sample was amended with 5 vol% of pyrogenic biochar (pine wood, PYREG® GmbH, Dörth/Germany) with a particle size <0.1 mm and

homogenized. Subsequently, the biochar soil was sterilized with 40.000 Gy using a Cobalt-60 $\gamma$-radiation source and an exposure time of 2 weeks. The resulting soil had a pH of 7.1 in 0.01M $CaCl_2$ solution, $C_{org}$ of 28.1 mg $g^{-1}$ as well as an oven-dry density of 1.36 g $cm^{-3}$ and a grain gross density of 2.54 g $cm^{-3}$ resulting in a pore volume of 46.4%.

To provide a soil-born microbial population, 1200 g of untreated fresh soil were extracted

with 1560 ml of 10-fold diluted modified R2A broth (0.1 g $l^{-1}$ $NH_4NO_3$, 0.05 g $l^{-1}$ yeast extract, 0.05 g $l^{-1}$ soy peptone, 0.05 g $l^{-1}$ casamino acids, 0.05 g $l^{-1}$ glucose, 0.05 g $l^{-1}$ soluble starch, 0.03 g $l^{-1}$ $K_2HPO_4$, 0.0024 g $l^{-1}$ $MgSO_4$, pH 7.2 ± 0.2, autoclaved at 121°C for 20 min) by end-over end shaking for 3 hours. The extract was filtered twice through two layers of laboratory tissue paper and afterwards split into two halves. One half was

autoclaved at 120°C for 20 minutes, whereas the other half remain untreated.

### 2.2 Incubation and sampling

Under sterile conditions, each 300 g soil sample were filled into two triplicates of pF-bioreactors diagrammed in Fig. 1. The reactors provide constant matrix potential and

sterile air supply for microbial soil containment experiments. With a flow rate of 0.4 l $min^{-1}$, the headspace was continually replaced by room air filtered with an 0.2 µm membrane filter. The hydrostatic head was 120 cm and provided a soil water content of about 35.0 vol % and a soil air content of 11.5 vol% at pF 2.1.

The first triplicate ($SP_{soil}$) was inoculated with 100 ml of the untreated soil extract to re-

establish the former microbial population, whereas the second triplicate got the sterilized inoculate to start as abiotic environment and be susceptible for infection by air-born microorganisms ($SP_{air}$). Soil extract exceeding the adjusted soil water content was removed by the hydrostatic head and discarded.

The soil columns were incubated for a total of 76 days at room temperature between

24.5°C and 32.5°C until day 50 and 8°C from day 51 to 76. In addition, hanging water columns were disconnected at day 24 and reconnected after add-on of 100 ml of 10-fold





diluted modified R2A broth at day 50. This resulted in a stress factor setting including warm-humid conditions from day 1 to 24, warm and drying-out conditions between day 25 and 50 as well as cold-humid conditions from day 51 to 76, that is supposed to be promoting biofilm production.

Soil sampling for DNA analysis was performed using sterile plastic pipes. A 500 mg composite sample at 3 evenly distributed sampling points was taken from each column 18 and 29 hours as well as 3, 5, 16, 49, 51 and 76 days after inoculation. The $SP_{soil}$ samples were taken in the cleanbench, whereas $SP_{air}$ was exposed to the unsterile room atmosphere for 15 min during sampling to enforce contamination. The samples were filled in 2 ml reaction tubes and stored at -20°C for later DNA extraction and quantification.

After day 76, soil parallels were removed from reactors and air-dried for 2 weeks in a laminar flow hood. A pH of 6.8 was measured for all variants. Afterwards soil aggregates of a size 0.63 to 2.0 mm were separated and used for analysis of aggregate stability.

## 2.3 DNA extraction and qPCR

DNA was extracted from 370 mg dry soil equivalent using a NucleoSpin® Soil Kit (MACHEREY-NAGEL GmbH & Co. KG, Düren/Germany) following the manual instructions. DNA sample purity, represented by 260/280nm extinction ratio, was determined with a NanoDrop1000 spectrophotometer (NanoDrop Products, Wilmington, DE, USA) and assessed as free of contamination.

For quantification of different phylogenetic classes (acidobacteria, actinobacteria, α- and β-proteobacteria) and domains (archaea, eubacteria, fungi), a quantitative real-time PCR with specific primer pairs (Table 1) was performed using a QuantStudio™ 12K Flex Real-Time PCR System (Life Technologies, Grand Island, NY/USA). The reaction mix per sample contained 4 µl of 5x HOT FIREPol® EvaGreen® HRM Mix ROX (Solis Biodyne, Tartu/Estonia), each 0.25 µl of the proper 10pM fwd and rev primer solution (biomers.net, Ulm, Germany; Table 2), 14.5 µl of PCR $H_2O$ and 1 µl of template DNA. Amplification of DNA templates was executed having an initial denaturation at 95°C for 15 min followed by 40 thermocycles consisting of a denaturation at 95°C for 15s, annealing for 20s at primer-specific temperatures listed in Table 1 and elongation at 72°C for 30s. PCR was checked for consistency by melting curve analysis.

Extracted DNA from standard organisms named in Table 1 was used as DNA standard, whereas DNA of non-target organisms from soil samples in return functioned as negative control. Sample-DNA dilution ranged between 1:1 and 1:100.










### 2.4 Aggregate stability

Soil aggregate stability was determined using successive ultrasonication and density fractionation, separation and C/N-analysis of the released and floated POM (Kaiser and Berhe, 2014). Therefore, in a first step, 75 ml of 1.6 g $cm^{-3}$ dense sodium polytungstate

solution (SPT) were added to 15 g of air-dried $SP_{soil}$ and $SP_{air}$ soil samples. Also a third triplicate ($SP_{pure}$) – remained unincubated – was analyzed. After 30 min of SPT infiltration into the soil matrix, free light fraction (fLF) floating after centrifugation at 3,569 G for 26 min was separated by filtering the SPT solution through an 1.5 μm pore size glass fibre filter. In a second step, the remaining soil was filled up to 75 ml SPT solution, ultrasonicated with

50 J $ml^{-1}$ using an ultrasonication device (Branson© Sonifier 250), centrifuged and floating occluded light fraction ($oLF_{50}$) was separated. The energy output of the ultrasonication device was therefor determined by measuring the heating rate of water inside a dewar vessel (Schmidt et al., 1999). Again the SPT solution was filled up to 75 ml. The sample was treated with an additional energy of 450 J m $l^{-1}$ and centrifuged. Floated occluded light

fraction ($oLF_{500}$) and heavy fraction (HF) were separated and all separated light fractions (LFs) and HF samples were frozen at 20°C, lyophilized, ground, dried at 105°C and analysed for organic carbon concentration using an Elementar Vario EL III CNS Analyzer. For comparison with natural-born soil aggregates, data of a soil sample ($A_{pure}$) mainly containing aggregates of a size between 0.63 and 2.0 mm with a $pH_{CaCl2}$ of 6.9, $C_{org}$ of 8.7

mg $g^{-1}$, a carbonate concentration of 0.2 mg $g^{-1}$ and a $C_{mic}$ of 352 μg $g^{-1}$ were included (Büks and Kaupenjohann, in revision). This soil has the same origin as SP soil samples.

### 2.5 Statistics

Statistic analysis of DNA data comprised calculation of mean values, standard deviations

and analysis of variance. For each sampling date class-, domain- and total DNA concentrations of $SP_{soil}$ and $SP_{air}$ parallels were tested for significant difference using a two-sample t-test. Therefore each variant was tested with Shapiro Wilk test and Levene test to provide normal distribution and evidence of variance homogeneity. Similarly, LF carbon releases of $SP_{soil}$, $SP_{air}$, $SP_{pure}$ and $A_{pure}$ were analyzed. In addition, a Tukey test

was applied.



## 3 Results

### 3.1 Microbial population analysis

The DNA extracted from soil samples shows qualitative differences in the composition of eubacterial populations and furthermore quantitative differences in the fungal population between $SP_{soil}$ and $SP_{air}$. It is expressed as ng DNA per mg dry soil (ng mg$^{-1}$) and includes intra- and extracellular DNA. (Fig. 1)

*Total eubacterial population is equal in both variants between day 49 and 76:* Through day 16, abundances of the total eubacterial population (DNA$_{EUB}$, Eub338/Eub518 primer pair) of $SP_{soil}$ and $SP_{air}$ differ significantly from each other ($p<0.05$): As the amount of eubacterial DNA in $SP_{soil}$ increases continuously from 1.28 ng mg$^{-1}$ at trial beginning to 16.27 ng mg$^{-1}$ at day 6 and subsequently decreases to 6.67 ng mg$^{-1}$ at day 76, DNA of $SP_{air}$ increases continuously from 0 ng mg$^{-1}$ at day 0 to 13.58 ng mg$^{-1}$ at day 49 and then decreases ultimately to 6.74 ng mg$^{-1}$. Between day 16 and 49 the significant difference of $SP_{soil}$ and $SP_{air}$ eubacterial abundances disappeared and the population density of both variants converges to highest similarity at day 76.

The sum of total measured DNA (DNA$_{tot}$=DNA$_{EUB}$+DNA$_{FUNG}$+DNA$_{ARCH}$) in $SP_{air}$ is quantified to around 2 ng mg$^{-1}$ until day 6, increases to 13.61 ng mg$^{-1}$ at day 49 and decreases again until 6.8 ng mg$^{-1}$ at day 76. $SP_{soil}$ increases from 2.38 ng mg$^{-1}$ at the beginning to 19.64 ng mg$^{-1}$ at day 6 and then decreases to an endpoint of 11.39 ng mg$^{-1}$. Except day 49 and 51 both variants show significant differences in DNA abundance, which is mainly due to fungal DNA.

*Fungi population only emerged in $SP_{soil}$:* Fungi (DNA$_{FUNG}$) show nearly no growth in $SP_{air}$ and remain at DNA concentrations below 0.2 ng mg$^{-1}$ for the whole experiment, whereas the fungal population of $SP_{soil}$ grows from 1.11 ng mg$^{-1}$ at day 0 to 5.56 at day 49 and ends at a value of 4.72 ng mg$^{-1}$. Despite high standard deviations, through day 6 fungal populations of $SP_{soil}$ and $SP_{air}$ differ significantly.

*Archaeal population is negligible:* Archaeal (DNA$_{ARCH}$) DNA amount is always >0,002 ng mg$^{-1}$ and does not show significant differences between variants.

*Eubacterial classes show significant differences between variants:* With look on eubacterial classes, acidobacteria show a high standard deviation, but differ significantly between variants at days 6, 16, 51 and 76. As $SP_{air}$ does not exceed values of 0.25 ng mg$^{-1}$, DNA concentration in $SP_{soil}$ growths from 0.44 ng mg$^{-1}$ at day 0 to 3.36 ng mg$^{-1}$ at day 16 and then stays between 2.19 an 3.22 ng mg$^{-1}$. Actinobacteria in $SP_{air}$ exhibit a nearly





constant DNA concentration between 0.22 and 0.31 ng mg$^{-1}$ until day 49, that does not exceed 0.52 ng mg$^{-1}$ afterwards. In contrast, the SP$_{soil}$ population rises from 0.33 ng mg$^{-1}$ at the beginning to 1.65 ng mg$^{-1}$ at day 6 and holds this value before decreasing to 0.95 ng mg$^{-1}$ between day 51 and day 76. Although SP$_{soil}$ shows an in tendency higher population, differences of both variants at days 51 and 76 are not significant. The development of SP$_{air}$ α-proteobacteria yielded in two plateaus: From day 0 to day 16 DNA concentration does not exceed 0.06 ng mg$^{-1}$ and remains between 0.16 and 0.21 ng mg$^{-1}$ from day 49 to day 76. The DNA concentration of SP$_{soil}$ increases from 0.07 ng mg$^{-1}$ at day 0 to 0.98 at day 6 and then decreases continuously to 0.39 ng mg$^{-1}$. At day 51 and 76 there are no significant differences between variants.

Among the examined eubacterial classes, only β-Proteobacteria show a higher population in SP$_{air}$ than in SP$_{soil}$: Through day 16 the DNA concentration in SP$_{air}$ remains smaller than 0.1 ng mg$^{-1}$, but afterwards strongly increases to 9.22 ng mg$^{-1}$ at day 49 followed by a decrease to 5.94 ng mg$^{-1}$ at the end. However, in SP$_{soil}$ DNA concentration rises from 0.22 ng mg$^{-1}$ to 2.84 ng mg$^{-1}$ until day 6, amounts 1.92 ng mg$^{-1}$ at day 16 and regulates itself to around 0.85 ng mg$^{-1}$ for days 49 to 76. Thereby, both variants differ significantly during the whole experiment.

*Variants are dominated by strongly differing microbial classes:* The percental relation of eubacterial class DNA to the total eubacterial DNA (Table 3) shows a dominance of acidobacterial DNA in SP$_{soil}$ having ratios between 23.02% and 36.77% at days 49 to 76, whereas values in SP$_{air}$ remain smaller than 0.86%. Also actinobacteria show a 3-fold higher percentage around 14.5% in SP$_{soil}$, compared to SP$_{air}$. In SP$_{air}$ and SP$_{soil}$, α-proteobacteria show percentages of 1.21% to 2.51% and 4.55% to 5.85%, respectively, and therefore do not represent a dominant class. In strong contrast, β-Proteobacteria hold increasing percentages from 67.89% to 88.10% in SP$_{air}$ compared to 8.67% to 12.27% in SP$_{soil}$. Cumulation show, that these classes cover 71.73% to 96.57% in SP$_{air}$, mainly dominated by β-proteobacteria, and 51.88% to 69.12% in SP$_{soil}$. Cumulative values show an increasing percentage over time in both variants.

## 3.2 Aggregate stability analysis

The relative net SOC release $C_{rel} = C_{frac} \cdot C_{\Sigma}^{-1}$, as defined as the ratio of the SOC release at the respective energy level ($C_{frac}$) to the cumulative SOC release of all separated LFs plus the sediment ($C_{\Sigma}$), shows no differences between SP$_{soil}$ and SP$_{air}$, but between the two and SP$_{pure}$. Data are shown as mean values and standard deviations of 3 parallels (Fig. 3):





SP$_{soil}$ and SP$_{air}$ do not differ in their relative fFL SOC release, which is around 4.6%. In contrast, the fLF release of SP$_{pure}$ amounts to 44.7%.

However, SP$_{soil}$, SP$_{air}$ and SP$_{pure}$ show differences of SOC release at 50 J ml$^{-1}$: SP$_{soil}$ releases 2.4% of its total particulate SOC in the oLF$_{50}$, which is significantly less than the A$_{pure}$ oLF$_{50}$ release of about 10.3%. SP$_{air}$ lies in between having 6.3%, without a significant difference to both.

At 500 J ml$^{-1}$, variants do not release different percentages of total particulate SOC. Whereas the oLF$_{500}$ detachment of SP$_{soil}$ and SP$_{air}$ are similar to each at 50 J ml$^{-1}$, SP$_{pure}$ is reduced to 1.3%. SP$_{air}$ show a tendency to exceed SP$_{soil}$ and SP$_{pure}$.

The SOC content of each sediment fraction corresponds to the sum of the respective LF SOC release and is 92.3% in SP$_{soil}$, 83.9% in SP$_{air}$ and 43.8 in SP$_{pure}$. Thus, only SP$_{pure}$ shows a significantly reduced SOC content remaining in the soil matrix.

As in consequence SP$_{soil}$ and SP$_{air}$ do not differ significantly in any fraction (although in tendency SP$_{air}$ releases more SOC than SP$_{soil}$ in both occluded light fractions), SP$_{pure}$ loses nearly half of its total particulate SOC in the fLF and additionally 10% after application of 50 J ml$^{-1}$.

The cumulative diagram of absolute SOC release (Fig. 4) identifies SP$_{soil}$ as the sample releasing the smallest amount of SOC over all LFs. SP$_{air}$ has an in tendency increased SOC release, whereas SP$_{pure}$ releases significantly more SOC. The SOC release of SP$_{soil}$ is identical to A$_{pure}$, although its SOC content is 3.2-times higher due to biochar amendment.










## 4 Discussion

Several studies demonstrated different compositions of EPS structural components in
single- and multi-species laboratory cultures. These variations are dependent on species
and incubation conditions. Most likely biofilms with fundamentally different microbial
population structures would have different chemical composition and/or shape linking soil
particles. Therefore we hypothesized that in a given soil a disparate development of the
microbial community would cause a significant difference in aggregate stability. This
hypothesis could be disproved for the given case and used best practice method, as
strongly diverging microbial populations did not affect aggregate stability significantly.

Evaluation of data took place based on a period of stable bacterial development within the
time span between day 49 and day 76: Showing a diverging development in the first 6
days, the amount of total eubacterial DNA in $SP_{soil}$ und $SP_{air}$ converge between day 6 and
day 49 leading to similar ($p<0.05$) quantities until the end of the experiment. Also, most of
the observed eubacterial classes in both variants seem to be established until day 49 and
show a stable or slightly decreasing population development in the final period from day 49
to day 76. (Fig. 1) These particular developments lead to a cumulative percental ratio of
acidobacteria, actinobacteria, α-proteobacteria and β-proteobacteria to total eubacterial
DNA, that increases from 71.7% to 96.6% in $SP_{air}$ and from 51.9% to 69.1% in $SP_{soil}$
between days 49 and 76. Therefore this bundle of eubacterial classes hold the majority in
both variants and become increasingly dominant. For these three reasons, the effect of
named eubacterial populations as well as fungi and archaea on aggregate stability is
discussed based on the final 28 days.

Although there is a similar total eubacterial DNA amount, population structure of examined
eubacterial classes is strongly varying between variants within the final period:
Acidobacteria and β-proteobacteria show a significant and actinobacteria an in tendency
but not significant difference between variants, whereas α-proteobacteria, which have low
abundances (< 6%) in both variants, did not develop differently between $SP_{soil}$ and $SP_{air}$.
Beside eubacteria, a fungal population developed in $SP_{soil}$, whose DNA spans 27.2% to
41.4% of the total measured population ($DNA_{tot}$) in the final period, whereas only very
small amounts of fungal DNA were found in $SP_{air}$ samples. Thus, fungal glomalin and
entanglement by fungal hyphae and filamentous bacteria (Aspiras et al., 1971; Gasperi-
Mago and Troeh, 1979; Tisdall, 1991; Bossuyt et al., 2001; Rillig et al., 2003) could
potentially affect aggregate stability only in $SP_{soil}$. In addition, no relevant archaeal





population was found in both variants, which also have capability to build biofilms (Fröls, 2013). Thus, ecosystems of both variants were dominated by different taxa: During the final period acidobacteria, actinobacteria and fungi together hold 60.5% to 71.3% of the total measured DNA in $SP_{soil}$. In contrast, $SP_{air}$ is nearly entirely dominated by β-proteobacteria, which provide 67.7% to 87.3% of the total measured DNA. We conclude, that in the final period both variants have strongly differing and each dominant microbial populations. This behavior implies a different development of EPS composition and biofilm structure. Based on our hypothesis, the different composition of microbial communities should have lead to a variation of aggregate stability between $SP_{soil}$ and $SP_{air}$.


Contrary to our forecast, in face of strongly differing microbial populations the incubated variants $SP_{soil}$ and $SP_{air}$ show no significant difference in aggregate stability in any fraction, although $SP_{soil}$ has a tendency to release less SOC. Even considering a relation of microbial development and aggregate stability in single parallels, no correlation of the growth/decline of a specific taxon and soil aggregate stabilization/destabilization was observed (data not shown). However, an increase of aggregate stability during the incubation experiment was demonstrated comparing incubated variants and $SP_{pure}$: The significant decrease of the net relative SOC content within the fLFs of $SP_{soil}$ and $SP_{air}$ points to an occlusion of POM and therefore soil aggregate formation during the incubation.



The absolute SOC release of $SP_{soil}$ and $SP_{air}$ is in the magnitude of natural formed aggregates from the original soil $A_{pure}$. That leads to the assumption, that the process of aggregation is at advanced stage after 76 days. Thus, breeding of stable soil aggregates in laboratory experiments is possible in short periods. Furthermore the absolute SOC release of both $SP_{soil}$ and $A_{pure}$ is identical, although their SOC content is 2.81% and 0.87%, respectively. That indicates an extensive occlusion of biochar within the $SP_{soil}$ matrix.



Following Büks and Kaupenjohann (in revision), biofilms play an important role in soil aggregate stabilization, whereas their microbial composition seems to be less relevant in this trial. What are the explanations?

First, relicts of the original EPS could endured drying, mechanical dispersion, γ-sterilization and recolonization along the whole soil treatment and then dominate or precure EPS induced aggregate stability. This causation for similar stabilities of $SP_{soil}$ and $SP_{air}$ seems improbable due to γ-degradation and metabolization (Kitamikado et al., 1990; Wasikiewicz et al., 2005), but could not be checked in this work.


More likely, the different microbial development in both incubated variants (1) causes only





a little difference in EPS molecular composition, that is not sufficient to affect aggregate stability, or (2) the span of possible molecular EPS compositions has in general no significant influence on the mechanical characteristics of biofilms.

Also physical stabilization of soil particles by fungal hyphae and glomalin have no significant influence on aggregate stability in this work, given that – in addition to a similar

eubacterial DNA amount in both variants – fungi represent 27% to 41% of the total measured DNA in $SP_{soil}$, but are nearly unrepresented in $SP_{air}$. However, the in tendency increased aggregate stability of $SP_{soil}$ at 50 and 500 J ml$^{-1}$ compared to $SP_{air}$ could underpin results of Molope et al. (1987) and Rillig (2004), that describe fungal hyphae and glomalin as factors of aggregate stability. A stabilizing effect of filamentous bacteria cannot

be examined due to similar representation of Acidobacteria in both variants and no further investigation of phylogenetic diversity within this group.

Probably, but also not tested, similar aggregate stabilities in both variants can be caused by a multi-species balancing mechanisms, in which a loss of biofilm coherence due to a dominance of one group of taxa is compensated by another group.

Thus, our findings show that soil-microbial ecosystems with vastly different community structures can develop nearly similar aggregate stability. It implies that even the occupation of  manifold niches by other taxa do not necessarily lead to a significant change in aggregate stability. Soil ecosystems could be able to compensate the influence of a population shift on EPS mechanical strength and aggregate coherence. Hence,

biofilms as an agent of soil aggregate stability could be rather able to compensate the influence of agricultural practice on soil microbial community. However, retention of aggregate stability after population shifts do not imply a retention of other soil microbial processes e.g. metabolical characteristics.

Even so it remains an open question, whether the use of a more accurate – still not

available – method of aggregate stability measurement could help to detect significant changes of aggregate stability, which are small but still have relevance for long term processes of SOC de-occlusion and soil erosion.




## 5 Conclusion

Our hypothesis could not be affirmed. After 76 days of incubation, two variants of the same sandy agricultural soil (Su3) established a similar eubacterial abundance, but different community structures – one strongly dominated by β-proteobacteria, the other one by acidobacteria, actinobacteria and fungi, that represent an additional DNA amount. Strong structural differences between these microbial communities did not cause significant differences in aggregate stability. Given that influence of biofilms on aggregate stability was proved, differences in soil microbial composition do not necessarily affect aggregate stability. This leads to assume that soil aggregate stability could be resilient in face of agricultural practices, that change microbial community structure. Nonetheless this practices can affect other soil metabolic characteristics by changing the microbial population. Therefore, the condition of the soil microbial community should be included in agricultural practice.

In addition, the incubation experiment demonstrated the possibility to breed stable soil aggregates in the laboratory within 3 month.

### *Author contribution*

The aggregate stability experiment was designed and carried out by F. Büks, the qPCR by P. Rebensburg. Data were evaluated by F. Büks with contributions from P. Lentzsch. The manuscript was prepared by F. Büks with contributions from M. Kaupenjohann.










## *Data availability*

**Alaoui, A. et al.**, 2011, doi:10.1016/j.still.2011.06.002

**Al-Halbouni, D. et al.**, 2009, doi:10.1016/j.watres.2008.10.008

**Allison, D. G.**, 1998, URL: https://tspace.library.utoronto.ca/handle/1807/82

**Aspiras, R. et al.**, 1971, URL: http://journals.lww.com/soilsci/Citation/1971/...

**Ayala-Hernández, I. et al.**, 2008, doi:10.1016/j.idairyj.2008.06.008

**Baldock, J.**, 2002, doi:0-471-60790-8

**Ball, B. and Robertson, E.**, 1994, doi:10.1016/0167-1987(94)90076-0

**Barthes, B. and Roose, E.**, 2002, doi:10.1016/S0341-8162(01)00180-1

**Baumgartl, T. and Horn, R.**, 1991, doi:10.1016/0167-1987(91)90088-F

**Béjar, V. et al.**, 1998, doi:10.1016/S0168-1656(98)00024-8

**Bengough, A. and Mullins, C.**, 1990, doi:10.1111/j.1365-2389.1990.tb00070.x

**Bennie, A. and Burger, R. d. T.**, 1988, doi:10.1080/02571862.1988.10634239

**Bossuyt, H. et al.**, 2001, doi:10.1016/S0929-1393(00)00116-5

**Bronick, C. J. and Lal, R.**, 2005, doi:doi:10.1016/j.geoderma.2004.03.005

**Büks, F. and Kaupenjohann, M.**, in revision, doi:10.5194/soil-2015-87

**Celik, G. Y. et al.**, 2008, doi:10.1016/j.carbpol.2007.11.021

**Chang, W.-S. et al.**, 2007, doi:10.1128/JB.00727-07

**Costerton, J. W. et al.**, 1995, URL: http://www.annualreviews.org/doi/pdf/10.1146/...

**Das, T. et al.**, 2014, doi:10.1371/journal.pone.0091935

**Davey, M. E. and O'toole, G. A.**, 2000, doi:10.1128/MMBR.64.4.847-867.20

**Fierer, N. et al.**, 2005, doi:10.1128/AEM.71.7.4117-4120.2

**Flemming, H.-C. and Wingender, J.**, 2010, doi:10.1038/nrmicro2415

**Fröls, S.**, 2013, doi:10.1042/BST20120304

**Gasperi-Mago, R. R. and Troeh, F. R.**, 1979, doi:10.2136/sssaj1979.03615995004300040029x

**Kaiser, M. and Berhe, A. A.**, 2014, doi:10.1002/jpln.201300339

**Kitamikado, M. et al.**, 1990, URL: http://aem.asm.org/content/56/9/2939.short

**Lane, D.**, 1991, URL: http://ci.nii.ac.jp/naid/10008470323/#cit

**Leigh, J. A. and Coplin, D. L.**, 1992, URL: http://www.annualreviews.org/doi/pdf/...

**Lueders, T. and Friedrich, M. W.**, 2003, doi:10.1128/AEM.69.1.320-326.2003

**Lützow, M. v. et al.**, 2006, doi:10.1111/j.1365-2389.2006.00809.x

**Mah, T.-F. C. and O'Toole, G. A.**, 2001, doi:10.1016/S0966-842X(00)01913-2

**Marty, N. et al.**, 1992, doi:10.1111/j.1574-6968.1992.tb05486.x



**Möhle, R. B. et al.**, 2007, doi:10.1002/bit.21448

**Molope, M. et al.**, 1987, doi:10.1111/j.1365-2389.1987.tb02124.x

**More, T. et al.**, 2014, doi:10.1016/j.jenvman.2014.05.010

**Muyzer, G. et al.**, 1993, URL: http://socrates.acadiau.ca/isme/Symposium16/muyzer.PDF

**Nunan, N. et al.**, 2003, doi:10.1016/S0168-6496(03)00027-8

**Overmann, J. et al.**, 1999, doi:10.1007/s002030050744

**Ozturk, S. and Aslim, B.**, 2010, doi:10.1007/s11356-009-0233-2

**Pal, A. and Paul, A.**, 2008, doi:10.1007/s12088-008-0006-5

**Ras, M. et al.**, 2011, doi:10.1016/j.watres.2010.11.021

**Rillig, M. C.**, 2004, doi:10.4141/S04-003

**Rillig, M. C. et al.**, 2003, doi:10.1016/S0038-0717(03)00185-8

**Roberson, E. B. and Firestone, M. K.**, 1992, URL:
http://aem.asm.org/content/58/4/1284.short

**Schmidt, M. et al.**, 1999, doi:10.1046/j.1365-2389.1999.00211.x

**Schmitt, J. and Flemming, H.-C.**, 1999, doi:10.1016/S0273-1223(99)00153-5

**Six, J. et al.**, 2002, doi:10.1023/A:1016125726789

**Stach, J. E. et al.**, 2003, doi:10.1046/j.1462-2920.2003.00483.x

**Steinberger, R. and Holden, P.**, 2005, doi:10.1128/AEM.71.9.5404-5410.2005

**Taylor, H. and Brar, G.**, 1991, doi:10.1016/0167-1987(91)90080-H

**Tisdall, J.**, 1991, doi:10.1071/SR9910729

**Van Loosdrecht, M. et al.**, 2002, doi:10.1023/A:1020527020464

**Votselko, S. et al.**, 1993, doi:10.1016/0167-7012(93)90016-B

**Wasikiewicz, J. M. et al.**, 2005, doi:10.1016/j.radphyschem.2004.09.021

**Weitere, M. et al.**, 2005, doi:10.1111/j.1462-2920.2005.00851.x

**Zhang, X. et al.**, 1998, doi:10.1016/S0273-1223(98)00127-9






## Acknowledgement


This project was financially supported by the Leibnitz-Gemeinschaft (SAW Pact for Research, SAW-2012-ATB-3). We are also grateful to our students Kathrein Fischer, Christine Hellerström, Annabelle Kallähne, Paula Nitsch, Susann-Elisabeth Schütze, Anne Timm and Karolin Woitke for pre-trials and soil preparation.



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



# *Tables*

**Table 1:** Target classes and domains, appropriate primer pairs, annealing temperatures (AT) and standard organisms for qPCR. (AWI=Alfred Wegener Institute, Helmholtz Centre for Polar and Marine Research; DSM=German Collection of Microorganisms and Cell Cultures; ZALF=Leibniz Center for Agricultural Landscape Research)

| Target organism | Primer pair | AT | Standard organism (origin) |
|---|---|---|---|
| Archaea | Ar109f / Ar915r | 57°C | *Methanosarcina mazei* (AWI) |
| Acidobacteria | Acido31 / Eub518 | 50°C | *Acidobacterium capsulatum* (DSM11244) |
| Actinobacteria | Actino235 / Eub518 | 60°C | *Streptomyces avermitis* (DSM46492) |
| α-Proteobacteria | Eub338 / Alf685 | 60°C | *Agrobacterium tumefaciens* pGV2260 (ZALF) |
| β-Proteobacteria | Eub338 / Bet680 | 60°C | *Burkholderia phymatum* (DSM17167) |
| Eubacteria | Eub338 / Eub518 | 53°C | *Pseudomonas putida F1* (ZALF) |
| Fungi | ITS1f / 5.8s | 52°C | *Verticillium dahliae* EP806 (ZALF) |





**Table 2:** Applied primer sequences for class- and domain-specific qPCR.

| Primer | Primer sequence | Reference |
| --- | --- | --- |
| 5.8s | 5'–CGCTGCGTTCTTCATCG–3' | (Fierer et al., 2005) |
| Acido31 | 5'–GATCCTGGCTCAGAATC–3' | (Fierer et al., 2005) |
| Actino235 | 5'–CGCGGCCTATCAGCTTGTTG–3' | (Stach et al., 2003) |
| Alf685 | 5'–TCTACGRATTTCACCYCTAC–3' | (Lane, 1991) |
| Ar109f | 5'–ACKGCTCAGTAACACGT–3' | (Lueders and Friedrich, 2003) |
| Ar915r | 5'–GTGCTCCCCCGCCAATTCCT–3' | (Lueders and Friedrich, 2003) |
| Bet680 | 5'–TCACTGCTACACGYG–3' | (Overmann et al., 1999) |
| Eub338 | 5'–ACTCCTACGGGAGGCAGCAG–3' | (Lane, 1991) |
| Eub518 | 5'–ATTACCGCGGCTGCTGG–3' | (Muyzer et al., 1993) |
| ITS1f | 5'–TCCGTAGGTGAACCTGCGG–3' | (Fierer et al., 2005) |



**Table 3:** Measured eubacterial class DNA of $SP_{air}$ and $SP_{soil}$ variant in relation to total eubacterial DNA in percent at days 49, 51 and 76.

| eubacterial class | $SP_{air}$ | | | $SP_{soil}$ | | |
|---|---|---|---|---|---|---|
| at day | 49 | 51 | 76 | 49 | 51 | 76 |
| Acidobacteria | 0.47 | 0.79 | 0.86 | 23.02 | 32.69 | 36.77 |
| Actinobacteria | 2.16 | 5.97 | 5.51 | 14.42 | 14.94 | 14.23 |
| α-Proteobacteria | 1.21 | 2.37 | 2.51 | 5.77 | 4.55 | 5.85 |
| β-Proteobacteria | 67.89 | 79.75 | 88.10 | 8.67 | 8.83 | 12.27 |
| *sum* | *71.73* | *88.88* | *96.57* | *51.88* | *60.88* | *69,12* |









## *Figures*

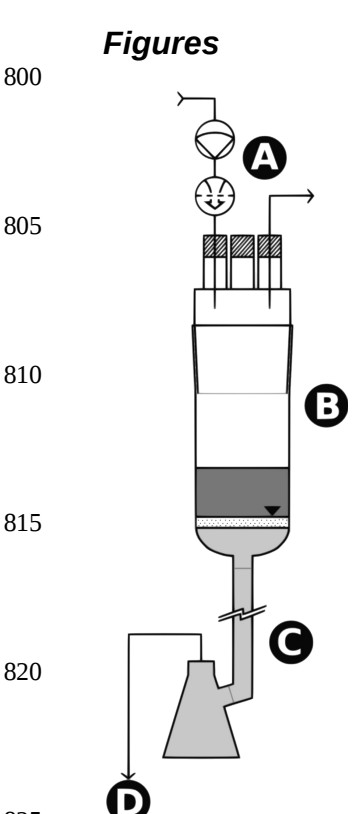

***Fig. 1:*** *Field capacity bioreactor with its components A) air supply composed of diaphragm pump and membrane filter, B) filter column with soil sample (dark grey) and filter plate (dotted), C) hydrostatic head (pale grey) and D) liquid waste container.*





**Fig. 2:** DNA concentrations of phylogenetic classes and domains in soil with natural inoculate (SP$_{soil}$) and air-born infection (SP$_{air}$) (values in ng DNA per mg dry soil)










**Fig. 3:** Relative SOC release of variants (SP$_{soil}$, SP$_{air}$, SP$_{pure}$) at different energy levels (0, 50, 500 J ml$^{-1}$), illustrated by Tukey test characters (a, ab, b).











**cumulative LF SOC release [mg/g]**

**Fig. 4:** Cumulative data of absolute SOC release (in mg SOC per g dry soil) of SP$_{soil}$, SP$_{air}$, SP$_{pure}$ and A$_{pure}$ as a function of applied energy. (*) marks A$_{pure}$ as measured at 0, 50 and 300 J ml$^{-1}$.