# Peer review of "Relation of aggregate stability and microbial diversity in an incubated sandy soil."

_SOIL, 2016_

## Referee Comment (RC1) · Anonymous Referee #1 · 22 Mar 2016

This study addressed a very interesting question and appears to be overall nicely executed and the manuscript is well written. My main concern with the study is inadequate number of replications, the lack of statistical significance that probably ensues from it, and inadequate treatment of that lack of statistical significance.

If I understand correctly, there were only three true replications of the studied systems? Given very high variability of soil aggregation data, no wonder that no statistically significant differences were observed. But the observed tendencies appear to be consistent with the authors' hypothesis. In such cases it is strongly recommended to conduct post-hoc power analysis to address the sufficiency of the replications and the size of the differences that could be statistically detected given the observed variability and the numbers of replications used. I would strongly recommend the authors to conduct such analysis.

Other questions: 1) The choice of microorganism sources: soil born - air-born. Does not it by definition biases things towards greater aggregation in soil-born case, because they will for sure have fungi? If the authors are truly after biofilms they should choose a more biofilm oriented set of sources.

2) It is not clear where the air-born microorganisms come from. Line 195 paragraph talks about sterile air supply and line 215 paragraph states that exposure to unsterile air was done after the incubation?

3) Description of statistical methods is not clear. There are two factors here - two soil treatments and several sampling times. Why this is not analyzed as a two factor experiment with repeated measures? There were only three replications analyzed - how the tests for normality and equal variance could be conducted with so few data points. What is meant by "variant"?

4) Results section can be shortened and a lot of things in Discussion should be moved to the Results. As of now the Results contain a lot of verbal descriptions of how numbers go up and down on the figures and this is not helpful. I would suggest to focus on bringing to reader's attention the key trends and points of interest instead (those are present in the Discussion and should be moved to the Results – e.g., the material in l. 405 paragraph).

5) Should show on the figures and in Table 3 when the differences are statistically significant and when they are not.

6) Ll 384-385 – unclear, please rewrite.

---

## Referee Comment (RC2) · Anonymous Referee #2 · 6 Apr 2016

General comments The authors intend to analyze the effects of microbial diversity on the formation and stability of aggregates in a sandy soil admixed with 5 vol

Some details: Title The title did not completely meet the content of the paper: Most of the discussion focusses on the effects of inoculation on soil microbial diversity and its possible effects on net and cumulative SOC release (may be used as a measure for aggregate stability, however a reference is missing). Additionally the manuscript focused on a mixture of soil plus 5volMay be the statement at L137 will reflect to a larger extent the content of the paper: "Testing the influence of two different microbial communities on aggregate stability in a sandy soil"?? Here the biochar effects should also be considered. Explain why not using pure soil but a mixture of soil plus biochar. Further I would like to ask the authors to consider the aspect that gamma radiation to free organic matter because of cell damage.

[Figure]

Abstract L43. Please correct into "Therefore samples of a… were…." L46 A none-incubated subsamples is used as a control. The abstract should also contain a statement reflecting the conclusion of your study.

Introduction L148 please add an explanation why your studied the effects of microbial communities of aggregate formation by using a soil + biochar mixture instead of a soil itself? Please note within the conclusion how this statement is tested: true or false?

Material and Methods Please add references on the methods used for -homogenization of biochar + soil mixture, - gamma radiation, -filtration and -autoclave procedure etc. Note where the "R2A broth" ("mixture" may be a more common synonym for broth, or is "R2A broth" a trade name?, in this case please mark it accordingly), add a reference. L191 Are there any references on these procedures please add them or add missing information (e.g. testing incubation conditions to be constant…) L202 Please explain why "soil extract could exceed the adjusted water content " (at line 194 you stated a "…constant matrix potential.."). Please explain the reason to discard these exceeds. L204ff please add references or explain the reasons for choosing the mentioned gradient in temperature, the disconnection of the hanging water columns etc. L221 Please add references on the procedures described here… L224 please add information how the 260/280nm ratio is obtained (I assume an UV-vis spectral analysis. However, this needs to be mentioned within the manuscript including a reference that shows this ratio to reflect purity of DNA samples). L225-240 Please add all missing references… L257 add a reference L263 please note a reference on the statistics. L275-335 PLEASE explain all abbreviations like DNAEUB (is it equal to EUB338/EUB518 primer pair?) and their meaning with respect to microbial diversity within the materials and methods section. …… L336 please add within materials and methods how you analyze the particle size fractions for Cfrac and Crel … Please explain the meaning of "oLF500…" etc. L379-384 seem to belong into introduction (include missing references)

If you introduce an abbreviation please use it consistently throughout the whole manuscript.

Please note within discussion and conclusion how the hypotheses (including the last statement of the introduction (L147-149) given in the introduction were tested. Add a summary of those results within the abstract.

L415-420 please clarify these statements. Please add a discussion on the effect of gamma radiation on the amount of decomposable organic matter... Such organic molecules may (i) interact with mineral soil as well as the biochar components in an abiotic way and may potentially force microbial activity.

Figs. mention within figure 1 where the soil sample is located?

Please explain which figure/table represents the data on aggregate stability? Are the data on "cumulative SOC release" used as a measure for aggregate stability? Explain why.
* * *

---

## Referee Comment (RC3) · Anonymous Referee #3 · 11 Apr 2016

[12pt]article natbib lineno graphicx enumerate

**General Comments**

1. The introduction to the discussion paper focusses too greatly on biofilms, EPS composition and formation and bacterial composition with little or no discussion of aggregate stability (the aim of the paper being to relate the former to the latter). Aggregate stability is determined by both biotic and abiotic factors and this should

be commented upon.

2. The authors use sonication to disperse aggregates and then measure the release of organic carbon (OC) as a measure of aggregate stability. I am not familiar with any studies which state that aggregate stability can be measured by the quantity of OC released. The authors refer to Kaiser & Berhe as the basis for their method, but in this paper Kaiser & Berhe do not state their approach is a means to measure aggregate stability. Aggregate stability is typically measured by successive reduction in particle size (typically mean weight diameter) of aggregates, not by reference to the quantities of OC released. If it were possible to show a strong linear relationship between aggregate size and OC released then it might be possible to infer aggregate stability, but I do not consider the current approach in the discussion paper to be a measure of aggregate stability. The authors need to justify their approach in the context of the published literature on aggregate stability.

3. The language and grammar used in the paper requires a considerable amount of revision before the paper could be accepted for publication. I have suggested several amendments in the technical corrections but there are many more than this.

3. I was not convinced by the evidence that biofilms are formed as a reaction to ecological stress - the citation referred does not relate to this. Please provide clear evidence/citation to this association.

4. What statistical significance can we place on results with only three replicates?

5. Line 219 - 'were separated' - how were the aggregates separated?

**Technical Corrections**

1. Correct spellings are: therefore, proteins,

2. use mineral, not inanimate

3. line 177; create, not receive

4. line 206; addition, not add-on

5. line 217; it is not clear what soil parallels are - please clarify

6. line 264; statistical analysis

7. line 340-341; it is not clear what is meant by 'but between the two and $SP_{pure}$',

8. line 480; Our hypothesis was not supported by the data.

---

## Editor Comment (EC1) · P.D. Hallett (Editor) · 3 May 2016

We now have comments from 3 referees on this paper. Some have raised concerns that will need to be addressed in a major revision and a strong response to their comments. There were concerns raised about the level of replication and statistical analysis of the data identified by all reviewers. We look forward to a revised submission of this manuscript.
* * *

---

## Author Comment (AC1) · 24 May 2016

Relation of aggregate stability and microbial diversity in an incubated sandy soil

Frederick Büks1, Philip Rebensburg2, Peter Lentzsch2 and M. Kaupenjohann1

1 Chair of Soil Science, Department of Ecology, Technische Universität Berlin. 2 Leibniz Center for Agricultural Landscape Research (ZALF e.V.), Müncheberg, Germany

Correspondence to: F. Büks (frederick.bueks@tu-berlin.de)

Final response to #Referee1

Dear Referee1, thank you very much for your important suggestions for the improvement of this work. In the following I present how to include them in a revised article.

1) The choice of microorganism sources: soil born - air-born. Does not it by definition biases things towards greater aggregation in soil-born case, because they will for sure have fungi? If the authors are truly after biofilms they should choose a more biofilm oriented set of sources. Our objective was to investigate the influence of truly different microbial populations on aggregate stability. Delmont et al. (2014) recently found that the development of microbial communities is controlled by physical-chemical properties of soils rather than start population: e.g. an initial population taken from a forest soil was given on a sterile grassland soil and there developed like the original grassland population. Therefore we were forced to be careful and chose two inoculates that definitely have no potential to converge their microbial abundances. The possibility to suppress fungi by antibiotics to see the pure bacterial effect, was discarded because of their potential effect on soil DOC and other microorganisms. Therefore, fungi are part of this biofilm. This information will be included in the revised paper.

2) It is not clear where the air-born microorganisms come from. Line 195 paragraph talks about sterile air supply and line 215 paragraph states that exposure to unsterile air was done after the incubation? "The incubation of both variants took place in columns with sterile air supply to get e.g. similar evaporation rates and hinder permanent infection. On the other hand, during sampling the air-born variant was exposed to room air to force infection, whereas the soil-born variant remained under sterile handling. After each sampling, both variants were reconnected to sterile air supply." We are going to explicate that in the revised article.

3) Description of statistical methods is not clear. There are two factors here – two soil treatments and several sampling times. Why this is not analyzed as a two factor experiment with repeated measures? There were only three replications analyzed - how the tests for normality and equal variance could be conducted with so few data points. What is meant by "variant"? AND If I understand correctly, there were only three true replications of the studied systems? Given very high variability of soil aggregation data, no wonder that no statistically significant differences were observed. But the observed

tendencies appear to be consistent with the authors' hypothesis. In such cases it is strongly recommended to conduct post-hoc power analysis to address the sufficiency of the replications and the size of the differences that could be statistically detected given the observed variability and the numbers of replications used. I would strongly recommend the authors to conduct such analysis. We restructured our statistic analysis in matters of your suggestions: "The statistic analysis of microbial populations and SOC release comprised calculation of mean values, standard deviations and analysis of variance. After application of the Shapiro-Wilk test (Shapiro and Wilk, 1965) samples were assumed to be normally distributed. T-test for unequal variances (Welch test) was used to test for significant differences of class, domain and total DNA concentrations between both incubated variants (SPsoil and SPair) (Ruxton, 2006). The Bonferroni correction was applied (Bland and Altmann, 1995). Total bacterial populations were assumed to be similar, if the difference is less than 1.2 ng DNA/ mg dry soil in the variants. SOC releases of SPsoil, SPair, SPpure and Apure were analyzed using one way ANOVA (Christensen, 1996)." The threshold of 1.2 ng DNA/ mg dry soil results in a narrowing of the "final period" from day 49-76 to day 51-76. This will be included to the manuscript. Power analyses is mostly used to estimate the needed sample size, to detect an assumed effect size with a given type 2 error rate. We didn't do a post-hoc power analysis for the following reasons: In case of the taxonomic comparison between the variants SPsoil and SPair significant differences in beta-proteobacteria, acidobacteria and fungi are given – therefore both variants are different and a type II error analysis is not necessary. The comparison of aggregate stability on the other hand showed no significant difference between variants. However, we forewent to apply a post-hoc power analysis in this case, since it only explains what is already known – the power of three parallels is low. Therefore we will add the following to the end of the discussion: "Our results give a first insight to the relation of microbial community composition, SOC release and aggregate stability. A more quantitative analysis would require more replicate samples, probably inclusion of soils from different land use and different microbial communities. However, this was beyond the scope of the present

study."

4) Results section can be shortened and a lot of things in Discussion should be moved to the Results. As of now the Results contain a lot of verbal descriptions of how numbers go up and down on the figures and this is not helpful. I would suggest to focus on bringing to reader's attention the key trends and points of interest instead (those are present in the Discussion and should be moved to the Results – e.g., the material in l. 405 paragraph). Thank you very much. I will do so after consulting the editors to best match their structural requirements.

5) Should show on the figures and in Table 3 when the differences are statistically significant and when they are not. I will include the level of significance to tables and figures.

6) Ll 384-385 – unclear, please rewrite. "Our results do not support this hypothesis, as strongly diverging microbial populations did not cause significantly different aggregate stabilities."

References

(Bland and Altmann, 1995) Bland, J. Martin, and Douglas G. Altman. "Multiple significance tests: the Bonferroni method." Bmj 310.6973 (1995): 170.

(Christensen, 1996) Christensen, Ronald. "One-way ANOVA." Plane Answers to Complex Questions. Springer New York, 1996. 79-93.

(Delmont et al., 2014) Delmont, Tom O., et al. "Microbial community development and unseen diversity recovery in inoculated sterile soil." Biology and Fertility of Soils 50.7 (2014): 1069-1076.

(Ruxton, 2006) Ruxton, Graeme D. "The unequal variance t-test is an underused alternative to Student's t-test and the Mann–Whitney U test." Behavioral Ecology 17.4 (2006): 688-690.

(Shapiro and Wilk, 1965) Shapiro, Samuel Sanford, and Martin B. Wilk. "An analysis of variance test for normality (complete samples)." Biometrika 52.3/4 (1965): 591-611.

Best regards, Frederick Büks

Please also note the supplement to this comment:
http://www.soil-discuss.net/soil-2016-14/soil-2016-14-AC1-supplement.pdf

---

## Author Comment (AC2) · 24 May 2016

Relation of aggregate stability and microbial diversity in an incubated sandy soil

Frederick Büks1, Philip Rebensburg2, Peter Lentzsch2 and M. Kaupenjohann1

1 Chair of Soil Science, Department of Ecology, Technische Universität Berlin. 2 Leibniz Center for Agricultural Landscape Research (ZALF e.V.), Müncheberg, Germany

Correspondence to: F. Büks (frederick.bueks@tu-berlin.de)

Final response to #Referee2

Dear Referee2. Thank you for your ideas for the improvement of this article and for your X-ray vision to find unreferenced details. In the following I want to answer your questions.

1) The title did not completely meet the content of the paper: Most of the discussion focusses on the effects of inoculation on soil microbial diversity and its possible effects on net and cumulative SOC release (may be used as a measure for aggregate stability, however a reference is missing). The method of ultrasonication/density fractionation invented by Golchin et al. (1994) is used as a blue-print for diverse surveys to analyze functional carbon Pools, e.g. in Cerli et al. (2012). As the dispersion cut-off (applied J/ml to release all occluded light fraction without destructive effects on mineral matrix and redistribution of SOM between fractions) strongly depends on soil properties, this method is not applicable to compare aggregate stability of different soils via SOC release. On the other hand, similar soils with similar mineralogy, SOC content, SOC distribution in fractions and binding patterns can be compared. Therefore discussion about the relation between SOC occlusion and aggregate stability will be included and the title is replaced by "Two different microbial communities did not cause differences in occlusion of POM and soil aggregate stability".

2) Additionally the manuscript focused on a mixture of soil plus 5vol. May be the statement at L137 will reflect to a larger extent the content of the paper: "Testing the influence of two different microbial communities on aggregate stability in a sandy soil"?? Here the biochar effects should also be considered. Explain why not using pure soil but a mixture of soil plus biochar. In the beginning, this experiment was part of a more extensive trial also including charcoal-free samples and further methods. Therefore, biochar is a relict. Lines 137-139 are replaced by "The aim of this work is to do a first step in this field by testing the influence of two fundamentally different microbial populations on the aggregate stability of a sandy agricultural soil containing 5 vol% biochar." Bacteria are colonizing biochar, and microbial community structure is affected by biochar (Jin, 2010). However, it is not the aim of this work to consider biochar effects and this is also not possible because of no biochar-free control.

3) Further I would like to ask the authors to consider the aspect that gamma radiation to free organic matter because of cell damage. Gamma radiation is surely damaging cells

and therefore enhancing soil DOC. However, the biochar soil was mixed before irradiation. It is distributed to the parallels of both incubated variants and SPpure afterwards. Therefore, we do not expect difference in DOC between variants in the beginning of the experiment.

4) Abstract L43. Please correct into "Therefore samples of a were." Thank you. Done.

5) L46 A none-incubated subsamples is used as a control. The abstract should also contain a statement reflecting the conclusion of your study. Thank you. Extension of the last sentence: "..., whereas comparison of incubated variants with a non-incubated sub-sample indicates strong stabilization during the incubation."

6) Introduction L148 please add an explanation why your studied the effects of microbial communities of aggregate formation by using a soil + biochar mixture instead of a soil itself? Please note within the conclusion how this statement is tested: true or false? See 2). Extension of the last sentence in line 148: "... because of their influence on aggregate stability."

7) Material and Methods Please add references on the methods used for - homogenization of biochar + soil mixture, - gamma radiation, -filtration and -autoclave procedure etc. We will add the following text to the manuscript: "Homogenization took place by over-head-shaking for approximately 1 minute. Gamma-irradiation was performed following McNamara et al. (2003) using additional 20 kGy. We used a 1.5 $\mu$m pore size glass fibre filter for adequate separation of POM, although 0.45 $\mu$m are required to retain single bacterial cells. And the autoclave program is typical for sterilization of liquid and solid media (Atlas, 2010)."

8) Note where the "R2A broth" ("mixture" may be a more common synonym for broth, or is "R2A broth" a trade name?, in this case please mark it accordingly), add a reference. To our knowledge "R2A broth" is a very common term. The reference for the composition of R2A is Atlas (2010).

9) L191 Are there any references on these procedures please add them or add missing information (e.g. testing incubation conditions to be constant) No references. In a pre-trial we add 100 ml of tap water to 300 g of soil sample. Impounded water was rejected within 15 minutes. The adjustment to 37 vol% soil water content at pF=2 took place within 4 days. This information will be added to the text.

10) L202 Please explain why "soil extract could exceed the adjusted water content " (at line 194 you stated a "constant matrix potential.."). Please explain the reason to discard these exceeds. "The soil has a bulk density of 1.36 g/cm$^3$. A water content of 35 vol% equates to 77 ml. Giving 100 ml soil extract to the sample, 23 ml are subsequently removed by the hydrostatic head when water capacity is adjusted to pF=2.1." This will be added to the manuscript.

11) L204ff please add references or explain the reasons for choosing the mentioned gradient in temperature, the disconnection of the hanging water columns etc. As drought stress is one factor inducing biofilm buildup (see introduction), decrease of soil water content is mentioned to raise EPS production. Therefore the water columns were disconnected to facilitate drying. Although there are only a few studies about the influence of temperature on biofilm production, e.g. Di Bonaventura et al. (2008) points to increasing biofilm production with increasing temperature. Reduction of summer time incubation temperature to 8°C was mentioned to simulate soil temperatures in the early spring and autumn. Low temperatures give an advantage of fungal compared to bacterial growth (Borowik and Wyszkowska, 2015) and therefore have influence mainly on the domain composition of SPsoil samples. This information will be included to the manuscript.

12) L221 Please add references on the procedures described here It is already referenced as NucleoSpin® Soil Kit (MACHEREY-NAGEL GmbH & Co. KG, Düren/Germany

13) L224 please add information how the 260/280nm ratio is obtained (I assume an

UV-vis spectral analysis. However, this needs to be mentioned within the manuscript including a reference that shows this ratio to reflect purity of DNA samples). "260/230 nm and" added to the sentence: "DNA sample purity, represented by 260/230 nm and 260/280nm extinction ratio, was determined with a NanoDrop1000 spectrophotometer (NanoDrop Products, Wilmington, DE, USA) and assessed as free of contamination." The desired value for the 260/280nm extinction ratio is 1.80 (our mean value is 1.83, min=1.78, max=1.88), for 260/230 nm it is 2.0 (our mean is 1.80, min=1.68, max=1.92). (TECHNICAL BULLETIN NanoDrop)

14) L225-240 Please add all missing references References for Primers are listed in Table 2 following Fierer et al. (2005). Devices are already referenced in the text. Melting curve analysis is implemented in the QuantStudio™ 12K Flex Real-Time PCR System (Life Technologies, Grand Island, NY/USA).

15) L257 add a reference The Device (Elementar Vario EL III CNS Analyzer) is referenced in the text.

16) L263 please note a reference on the statistics. Shapiro-Wilk test: (Shapiro and Wilk, 1965) Welch test: (Welch, 1947) Bonferroni correction: (Bonferroni, 1936) one way ANOVA: (Christensen, 1996)

17) L275-335 PLEASE explain all abbreviations like DNAEUB (is it equal to EUB338/EUB518 primer pair?) and their meaning with respect to microbial diversity within the materials and methods section. Sorry for the imprecise description: DNAEUB = total eubacterial population amplified by Eub338/Eub518 primer pairs. All other abbreviations are already explained in the text. Archaea, eubacteria and fungi represent the 3 domains involved in soil aggregation. Fungi are known to stabilize soil aggregates, whereas influence of bacteria is shown to be minor (Tang et al., 2011). Acidobacteria, actinobacteria, $\alpha$- and $\beta$-proteobacteria are taxa commonly used in soil ecotyping (e.g. Jangid et al., 2011).

18) L336 please add within materials and methods how you analyze the particle size

fractions for Cfrac and Crel Please explain the meaning of "oLF500" etc. All this is already explained in lines 253-275.

19) L379-384 seem to belong into introduction (include missing references) If you introduce an abbreviation please use it consistently throughout the whole manuscript. All this is described in the introduction and only summarized as a liftoff at the beginning of the discussion. "EPS" is introduced in the introduction section and used consistently as the abbreviation of "extracellular polymeric substance" (the extracellular matrix of biofilms).

20) Please note within discussion and conclusion how the hypotheses (including the last statement of the introduction (L147-149) given in the introduction were tested. Add a summary of those results within the abstract. It's all part of the abstract and the discussion. The statement in lines 147-149 is related to lines 132-133 (introduction) and taken up in lines 460-468 (discussion). However, we do not think that methods should be repeated in the conclusion section – but that may depend on the philosophy of the writer.

21) L415-420 please clarify these statements. Thank you very much. I have really no idea what happened to this sentence :). New version: "We conclude, that both variants strongly differ in their community structure within the final period from day 49 to day 76, at which both communities are dominated by specific taxa. This development implies different EPS compositions and biofilm structures. Following the hypothesis of this work, the different composition of microbial communities should have lead to a variation of aggregate stability between SPsoil and SPair."

22) Please add a discussion on the effect of gamma radiation on the amount of decomposable organic matter Such organic molecules may (i) interact with mineral soil as well as the biochar components in an abiotic way and may potentially force microbial activity. Please see 3). Measuring and discussion of this effect is not the aim of this work.

23) Figs. mention within figure 1 where the soil sample is located? It is already mentioned in the caption of Figure 1 ("... soil sample(dark grey)...")

24) Please explain which figure/table represents the data on aggregate stability? Are the data on "cumulative SOC release" used as a measure for aggregate stability? Explain why. Data on "cumulative SOC release" generally show the SOC release as a function of applied mechanical force. Occluded POM, that is bond inside soil aggregates, is released after application of a certain level of mechanical force. A higher share of "weak" aggregates in soil will result in an increased release of SOC. Therefore the cumulative SOC release is used as a proxy of aggregate stability, but only if soils with similar amount and composition of SOC are compared. This relation will be presented in the discussion and is also part of the revised version of Büks and Kaupenjohann (in revision). See also 1). New caption of figure 3: "Relative SOC release of variants (SPsoil, SPair, SPpure) at different energy levels (0, 50, 500 J ml-1). Highest SOC release is associated with lowest aggregate stability at the respective energy level." New caption of figure 4: "Cumulative data of absolute SOC release (in mg SOC per g dry soil) of SPsoil, SPair, SPpure and Apure as a function of applied energy. (*) marks Apure as measured at 0, 50 and 300 J ml-1. Highest SOC release is associated with lowest aggregate stability."

References

(Atlas, 2010) Atlas, Ronald M. Handbook of microbiological media. CRC press, 2010.

(Bonferroni, 1936) Bonferroni, Carlo E. Teoria statistica delle classi e calcolo delle probabilita. Libreria internazionale Seeber, 1936.

(Borowik and Wyszkowska, 2015) Borowik, Agata, and Jadwiga Wyszkowska. "Impact of temperature on the biological properties of soil." International Agrophysics 30.1 (2016): 1-8.

(Büks and Kaupenjohann, in revision) Büks, F., and M. Kaupenjohann. "Enzymatic

biofilm detachment causes a loss of aggregate stability in a sandy soil." Soil Journal, in revision.

(Cerli et al., 2012) Cerli, C., et al. "Separation of light and heavy organic matter fractions in soil—Testing for proper density cut-off and dispersion level." Geoderma 170 (2012): 403-416.

(Christensen, 1996) Christensen, Ronald. "One-way ANOVA." Plane Answers to Complex Questions. Springer New York, 1996. 79-93.

(Di Bonaventura et al., 2008) Di Bonaventura, G., et al. "Influence of temperature on biofilm formation by Listeria monocytogenes on various food‐contact surfaces: relationship with motility and cell surface hydrophobicity." Journal of applied microbiology 104.6 (2008): 1552-1561.

(Fierer et al., 2005) Fierer, Noah, et al. "Assessment of soil microbial community structure by use of taxon-specific quantitative PCR assays." Applied and environmental microbiology 71.7 (2005): 4117-4120.

(Golchin et al., 1994) Golchin, A., et al. "Study of free and occluded particulate organic matter in soils by solid state 13C CP/MAS NMR spectroscopy and scanning electron microscopy." Soil Research 32.2 (1994): 285-309.

(Jangid et al., 2011) Jangid, Kamlesh, et al. "Land-use history has a stronger impact on soil microbial community composition than aboveground vegetation and soil properties." Soil Biology and Biochemistry 43.10 (2011): 2184-2193.

Jin, Hongyan. "Characterization of microbial life colonizing biochar and biochar-amended soils." (2010).

(McNamara et al., 2003) McNamara, N. P., et al. "Effects of acute gamma irradiation on chemical, physical and biological properties of soils." Applied Soil Ecology 24.2 (2003): 117-132.

(Shapiro and Wilk, 1965) Shapiro, Samuel Sanford, and Martin B. Wilk. "An analysis of variance test for normality (complete samples)." Biometrika 52.3/4 (1965): 591-611.

(Tang et al., 2011) Tang, Jia, et al. "Influence of biological aggregating agents associated with microbial population on soil aggregate stability." Applied Soil Ecology 47.3 (2011): 153-159.

(TECHNICAL BULLETIN NanoDrop) http://nanodrop.com/Library/T042-NanoDrop-Spectrophotometers-Nucleic-Acid-Purity-Ratios.pdf

(Welch, 1947) Welch, Bernard L. "The generalization ofstudent's' problem when several different population variances are involved." Biometrika 34.1/2 (1947): 28-35.

Best regards, Frederick Büks

Please also note the supplement to this comment:
http://www.soil-discuss.net/soil-2016-14/soil-2016-14-AC2-supplement.pdf

---

## Author Comment (AC3) · 24 May 2016

Relation of aggregate stability and microbial diversity in an incubated sandy soil

Frederick Büks1, Philip Rebensburg2, Peter Lentzsch2 and M. Kaupenjohann1

1 Chair of Soil Science, Department of Ecology, Technische Universität Berlin. 2 Leibniz Center for Agricultural Landscape Research (ZALF e.V.), Müncheberg, Germany

Correspondence to: F. Büks (frederick.bueks@tu-berlin.de)

Final response to #Referee3

Dear Referee3. Thank you very much for your suggestions to improve the quality of this article.

1) The introduction to the discussion paper focusses too greatly on biofilms, EPS composition and formation and bacterial composition with little or no discussion of aggregate stability (the aim of the paper being to relate the former to the latter). Aggregate stability is determined by both biotic and abiotic factors and this should be commented upon. In line 103 we add as follows: "... : EPS is one of manifold factors of soil aggregate stabilization. Permanent and variable charges of silicates, (hydr)oxides of Fe, Al and Mn, phosphates, carbonates as well as POM interact to each other meditated by multivalent cations with small hydrate shells (e.g. $Ca_{2+}$, $Fe_{3+}$ and $Al_{3+}$). Also dissolved organic matter (DOM) like humins, plant exudates and diverse decomposing products builds physico-chemical links between soil particles and covers mineral surfaces. In addition, fungal mycelia and fine roots form a stabilizing network in and around soil aggregates. (Jastrow and Miller, 1997; Bronick and Lal, 2005)". Also "However, the contribution of biofilms to this stabilization was not yet quantified." was added to line 429.

2) The authors use sonication to disperse aggregates and then measure the release of organic carbon (OC) as a measure of aggregate stability. I am not familiar with any studies which state that aggregate stability can be measured by the quantity of OC released. The authors refer to Kaiser & Berhe as the basis for their method, but in this paper Kaiser & Berhe do not state their approach is a means to measure aggregate stability. Aggregate stability is typically measured by successive reduction in particle size (typically mean weight diameter) of aggregates, not by reference to the quantities of OC released. If it were possible to show a strong linear relationship between aggregate size and OC released then it might be possible to infer aggregate stability, but I do not consider the current approach in the discussion paper to be a measure of aggregate stability. The authors need to justify their approach in the context of the published literature on aggregate stability. We will include the following statement in line 420 before discussion of SOC release. "Generally, aggregate stability is characterized by determining the reduction in aggregate size after application of mechanical force. The commonly used methods are dry and wet sieving. However, the destruction of

soil aggregates by ultrasonication has an advantage over these methods, which is the quantification of the applied energy (North, 1976). It is used for studying reduction of aggregate size (Imeson and Vis, 1984) as well as detachment of occluded POM carbon (Golchin et al., 1994). Kaiser and Berhe (2014) reviewed 15 studies using ultrasonication of soil aggregates in consideration of its destructiveness to the soil mineral matrix and occluded POM. They found destruction of POM at applied energy levels >60 J/ml, destruction of sand-sized primary particles at >710 J/ml and of smaller mineral particles at higher energy levels. We used this method of gentle POM detachment from soil aggregates to measure the oLF carbon release as a result of mechanical force and linked it to aggregate stability. Since Cerli et al. (2012) showed that the release of free and occluded light fractions strongly depends on soil properties like mineralogy, POM content and composition, this method is restricted to comparison of soils differing in none of these properties."

3) The language and grammar used in the paper requires a considerable amount of revision before the paper could be accepted for publication. I have suggested several amendments in the technical corrections but there are many more than this. Thank you very much. We will do our best and consult a native speaker. 1. Correct spellings are: therefore, proteins: Done 2. use mineral, not inanimate: Done. 3. line 177; create, not receive: Done. 4. line 206; addition, not add-on: Done. 5. line 217; it is not clear what soil parallels are - please clarify: "soil samples" 6. line 264; statistical analysis: Done. 7. line 340-341; it is not clear what is meant by 'but between the two and SP pure: "... whereas both differ significantly to SPpure." 8. line 480; Our hypothesis was not supported by the data: Done.

4) I was not convinced by the evidence that biofilms are formed as a reaction to ecological stress - the citation referred does not relate to this. Please provide clear evidence/citation to this association. Flemming and Wingender (2010) only refers to "..., but also act as genetic cross-over hotspot and collective digestive system for diverse soil nutrients." References for the evidence of reaction on ecological stress are given

in lines 73-74 (Roberson and Firestone, 1992; Mah and O'Toole, 2001; Weitere et al., 2005; Chang et al., 2007; Ozturk and Aslim, 2010) and will be included in this paragraph in line 119 ff. to avoid misunderstanding.

5) What statistical significance can we place on results with only three replicates? A more quantitative analysis would require more replicate samples. We will add the following to the end of the discussion: "Our results give a first insight to the relation of microbial community composition, SOC release and aggregate stability. A more quantitative analysis would require more replicate samples, probably inclusion of soils from different land use and different microbial communities. However, this was beyond the scope of the present study."

6) Line 219 - 'were separated' - how were the aggregates separated? Thank you. I will delete "... separated and ..."

References

(Bronick and Lal, 2005) Bronick, Carol Jean, and Rattan Lal. "Soil structure and management: a review." Geoderma 124.1 (2005): 3-22.

(Cerli et al., 2012) Cerli, C., et al. "Separation of light and heavy organic matter fractions in soil—Testing for proper density cut-off and dispersion level." Geoderma 170 (2012): 403-416.

(Chang et al., 2007) Chang, Woo-Suk, et al. "Alginate production by Pseudomonas putida creates a hydrated microenvironment and contributes to biofilm architecture and stress tolerance under water-limiting conditions." Journal of bacteriology 189.22 (2007): 8290-8299.

(Flemming and Wingender, 2010) Flemming, Hans-Curt, and Jost Wingender. "The biofilm matrix." Nature Reviews Microbiology 8.9 (2010): 623-633.

(Golchin et al., 1994) Golchin, A., et al. "Study of free and occluded particulate organic matter in soils by solid state 13C CP/MAS NMR spectroscopy and scanning electron

microscopy." Soil Research 32.2 (1994): 285-309.

(Imeson and Vis, 1984) Imeson, A. C., and M. Vis. "Assessing soil aggregate stability by water-drop impact and ultrasonic dispersion." Geoderma 34.3-4 (1984): 185-200.

(Jastrow and Miller, 1997) Jastrow, J. D., and R. M. Miller. "Soil aggregate stabilization and carbon sequestration: feedbacks through organomineral associations." Soil processes and the carbon cycle (1997): 207-223.

(Kaiser and Berhe, 2014) Kaiser, Michael, and Asmeret Asefaw Berhe. "How does sonication affect the mineral and organic constituents of soil aggregates?—A review." Journal of Plant Nutrition and Soil Science 177.4 (2014): 479-495. Mah, Thien-Fah C., and George A. O'Toole. "Mechanisms of biofilm resistance to antimicrobial agents." Trends in microbiology 9.1 (2001): 34-39.

(North, 1976) North, P. F. "Towards an absolute measurement of soil structural stability using ultrasound." Journal of Soil Science 27.4 (1976): 451-459.

(Ozturk and Aslim, 2010) Ozturk, Sahlan, and Belma Aslim. "Modification of exopolysaccharide composition and production by three cyanobacterial isolates under salt stress." Environmental Science and Pollution Research 17.3 (2010): 595-602.

(Roberson et al., 1992) Roberson, Emily B., and Mary K. Firestone. "Relationship between desiccation and exopolysaccharide production in a soil Pseudomonas sp." Applied and Environmental Microbiology 58.4 (1992): 1284-1291.

(Weitere et al., 2005) Weitere, Markus, et al. "Grazing resistance of Pseudomonas aeruginosa biofilms depends on type of protective mechanism, developmental stage and protozoan feeding mode." Environmental Microbiology 7.10 (2005): 1593-1601.

Best regards, Frederick Büks

Please also note the supplement to this comment:
http://www.soil-discuss.net/soil-2016-14/soil-2016-14-AC3-supplement.pdf